# Faculties to Support General Practitioners Working Rurally at Broader Scope: A National Cross-Sectional Study of Their Value

**DOI:** 10.3390/ijerph17134652

**Published:** 2020-06-28

**Authors:** Matthew R. McGrail, Belinda G. O’Sullivan

**Affiliations:** Rural Clinical School, The University of Queensland, Rockhampton 4700, Australia; belinda.osullivan@uq.edu.au

**Keywords:** general practitioners, postgraduate medical training, rural workforce, medical faculty, advanced skills, scope of practice, vocational education, primary health care, rural population, family physicians

## Abstract

Strategies are urgently needed to foster rural general practitioners (GPs) with the skills and professional support required to adequately address healthcare needs in smaller, often isolated communities. Australia has uniquely developed two national-scale faculties that target rural practice: the Fellowship in Advanced Rural General Practice (FARGP) and the Fellowship of the Australian College of Rural and Remote Medicine (FACRRM). This study evaluates the benefit of rural faculties for supporting GPs practicing rurally and at a broader scope. Data came from an annual national survey of Australian doctors from 2008 and 2017, providing a cross-sectional design. Work location (rurality) and scope of practice were compared between FACRRM and FARGP members, as well as standard non-members. FACRRMs mostly worked rurally (75–84%, odds ratio (OR) 8.7, 5.8–13.1), including in smaller rural communities (<15,000 population) (41–54%, OR 3.5, 2.3–5.3). FARGPs also mostly worked in rural communities (56–67%, OR 4.2, 2.2–7.8), but fewer in smaller communities (25–41%, OR 1.1, 0.5–2.5). Both FACRRMs and FARGPs were more likely to use advanced skills, especially procedural skills. GPs with fellowship of a rural faculty were associated with significantly improved geographic distribution and expanded scope, compared with standard GPs. Given their strong outcomes, expanding rural faculties is likely to be a critical strategy to building and sustaining a general practice workforce that meets the needs of rural communities.

## 1. Introduction

Rural communities worldwide need a sustainable, skilled medical workforce, especially general practitioners (GPs) and family physicians because they cover a wide range of primary and preventative healthcare needs for people in rural and isolated communities [1,2]. Universally, countries have sought to grow the rural GP workforce including in smaller rural communities because it provides essential services that mitigate the need for people to travel long distances for healthcare [3,4]. In response, many medical schools are aiming to enroll more rural background students and provide rural immersion experiences, which has shown positive results for rural work outcomes [5,6]. However, there is little evidence about national-scale interventions related to postgraduate education that supports targeting rural GP capacity, despite global recommendations identifying that tailored professional development improves the supply and retention of rural doctors [3]. 

Rural GPs are largely supported by mainstream faculties, but on their own these may provide limited attention to the skills and professional support needed by rural GPs. In response, many countries are developing specific postgraduate training and professional support pathways aimed to grow and support the skills that doctors need in rural practice, especially in primary care [7]. Australia is a unique case study of a country that developed two national faculties for rural GPs in the late 1990s: the Fellowship in Advanced Rural General Practice (FARGP) and the Fellowship of the Australian College of Rural and Remote Medicine (FACRRM), both of which promote more targeted education and continuous learning (Table 1) [8]. Australia’s rural faculties are the most developed internationally, but to date there is limited objective research evaluating their outcomes [9]. This includes whether they relate to practicing as GPs in rural communities (particularly smaller and isolated populations) and across a broader ‘rural generalist’ scope of services (doctors providing both comprehensive primary care and additional specialist services such as emergency medicine) [9,10]. Such evidence has the potential to inform the value of rural faculties and advise other countries seeking to implement similar strategies, including large-scale rural-centric vocational training (or residencies) and related professional development programs. Thus, our study aimed to evaluate the benefit of rural faculties for supporting a more geographically distributed rural GP workforce, practicing at broader scope.

The unique demands of working in rural communities and sometimes in isolated practice underpin the philosophy that rural GPs require tailored skills training, as well as ongoing professional development and networking opportunities. Rural faculties aim to create a community of practice that reduces professional isolation and increases doctors’ professional confidence and capabilities for providing a safe and high-quality breadth of care for rural communities [10,11,12,13]. The ongoing professional development accounts for the fact that the range of skills needed is not static, but evolves as the community profile changes, doctor’s interests develop, or specific healthcare needs change as doctors move between communities [14]. Maintaining both general and specialized skills relative to the specific needs of any one rural community underpins access to safe, life-saving medical interventions. 

Rural faculties that target the education and ongoing support for rural doctors may serve a particular role [15,16,17]. They can both generate a specifically skilled general practice workforce, while also addressing the need for GPs to access regular, rural-tailored medical education, professional networking, and support options [18]. Over mainstream faculties, they also enable learning that is based and contextualized in rural places, thereby minimizing travel and assisting real-world application. As such, rural faculties may be an important intervention for achieving a sustainable and high-quality medical services for rural communities [9,19,20,21]. 

Australia’s two national-scale rural faculties were developed at slightly different times (Table 1). Moreover, they involve somewhat different training elements, but each target relevant education to working in rural contexts, across a wider practice scope (Table 2) [22,23,24,25]. Firstly, embedded within the existing standard general practice training and fellowship of the Royal Australian College of General Practice (FRACGP) is the FARGP, which is associated with education and support of advanced skills in areas like emergency, obstetrics, anesthetics, and basic surgery. Secondly is a standalone and independent rural faculty of the Australian College of Rural and Remote Medicine (ACRRM), which enables a fellowship (FACRRM) with a core mission of developing and supporting rural doctors through education mainly based in rural areas, and is associated with a wider range of emergency skills, additional advanced skills, and experience in smaller and isolated communities. Table 1 summarizes the reasoning for each faculty’s emergence, both of which are equivalent for Australian Medical Council accreditation purposes. However, despite their potential value, evidence about rural faculties remains largely descriptive with limited evaluation of their workforce outcomes against mainstream approaches [7,26,27,28], including limited evidence from small scale rural residencies in other countries and other more localized postgraduate workforce interventions [29,30,31]. 

**Table 1 ijerph-17-04652-t001:** Timeline of the development of Australia’s two rural general practice faculties (the Fellowship in Advanced Rural General Practice (FARGP) and the Fellowship of the Australian College of Rural and Remote Medicine (FACRRM).

Year	Faculty Development Outcome	Related Information
1973	RACGP’s education program began (three years duration, end point FRACGP), but was not compulsory until 1996 [32].	
1989–1995	Existing GPs could take up a ‘grandfathering’ option (recognizing prior learning, RPL) for FRACGP [25].	Other existing doctors chose to have a formal ‘fellowship’, with no major implications to their practice.
1992	RACGP established a Faculty of Rural Medicine (FRM), recognizing the specific skills related to working in rural primary care.	This was the first acknowledgement that additional skills were required by GPs working in rural areas.
>1992	An optional Graduate Diploma of Rural General Practice (GradDip RurGP) was initiated, involving an additional year of training, with early results finding 70% retention in rural areas [33].	However, debate continued within the FRM and Rural Doctors Association of Australia (RDAA) if a full fellowship better recognized the standard of rural-specific learning.
1995–1996	A formal plebiscite led by the RDAA, asked rural doctors whether to continue in their academic association with FRM, whereby the majority voted to split from RACGP [22].	
1997	An independent rural-focused GP training college was initiated called the ACRRM, with a specific mission to deliver rural general practice training to the level of a fellowship.	This split of general practice training through two specialty college pathways remains to this day.
1998–1999	Approximately 700 rural-based GPs were ‘grandfathered’ (full RPL) ACRRM’s fellowship, as part of growing the rural supervisory faculty.	
2000	ACRRM commenced intake of new trainees, with training structured very similar to the modern 4-year qualification as per Table 2.	ACRRM also developed a rural-specific professional development and support program for existing members [22,24,33].
2006	RACGP’s FRM continued with its GradDip RurGP, transferring to a fellowship (FARGP), as per Table 2.	
	**Other key parallel interventions**	
1999–now	National policy (Rural Clinical Schools) supporting delivery of partial and full rural medical education programs [34,35].	
2000–now	National policy: 50% of general practice training occurs in rural areas.	
2007–now	Additionally, separate formal rural generalist (RG) pathways begun in various forms.	Queensland’s program (articulating with FACRRM and FARGP qualifications) linked to specific state-based awards recognizing and remunerating RG doctors [36,37].
2017–now	An inaugural Office of the National Rural Health Commissioner designed a scaled-up national RG pathway, with both FACRRM and FARGP agreed as the recognized RG doctor qualification [36].	

FACRRM: Fellowship of the Australian College of Rural and Remote Medicine; FARGP: Fellowship in Advanced Rural General Practice; RACGP: Royal Australian College of General Practitioners; ACRRM: Australian College of Rural and Remote Medicine.

## 2. Materials and Methods

This study used 2008–2017 data (waves 1–10) from the “Medicine in Australia: Balancing Employment and Life (MABEL)” study. MABEL collected annual cross-sectional survey data from a national panel of doctors across all career stages. It commenced in 2008, with 10,498 doctors (19% of the sampling frame, minimal participation bias) completing the initial survey (wave 1) [39]. There has subsequently been an annual 70–80% study retention rate, with new doctors topping up the sampling frame. MABEL was approved by the University of Melbourne Faculty of Business and Economics Human Ethics Advisory Group (Ref. 0709559) and the Monash University Standing Committee on Ethics in Research Involving Humans (Ref. CF07/1102-2007000291).

This study only included data from clinically active GPs and excluded those currently enrolled in vocational training (equivalent to ‘residency’) programs. Qualification data were self-reported across all 10 waves, responding to “What GP and other specialist postgraduate qualifications have you obtained in Australia? (e.g., FRACGP, FRACP, FACRRM, diploma)” and “Please list any GP and other medical qualifications you have obtained in Australia since the last time you completed the MABEL Survey”. Doctors were categorized into qualification categories, as described in the analysis below. 

Geographic distribution of the main work location was the primary outcome. Rurality was defined using the Department of Health’s Modified Monash Model (MMM) classification as metropolitan (MMM-1) rural (MMM 2–7) [40]. Some analyses further collapsed the rural category into MMM 2–3 (large rural/regional, >15,000 population) or MMM 4–7 (smaller rural <15,000 population or remote/frontier areas). Distribution was additionally explored by the state, due to the potential for state-based variation from both geography and state-based rural generalist support. Other key demographic factors were gender, age (<50, 50+), childhood background (at least 6 childhood years in a rural area), and place of basic medical training (Australian or New Zealand medical graduate (AMG), or overseas trained doctor (OTD)).

Measures of scope of practice were firstly defined by advanced skills area, whereby all doctors reported doing ”specialized training of at least 6 months which is outside the normal scope of practice for GPs”. Four groups were defined (see skills listed in Table 2): (i) practicing at least one additional skill; (ii) practicing one of the four procedural skills; (iii) having trained in an additional skill area, but not currently practicing it; and (iv) having trained in a procedural skill, but not currently practicing it. The latter two categories aimed to identify skill maintenance. These scope data were only available in Wave 10 (2017) of the MABEL survey. Secondly, scope was defined by a series of other indicators: work in a hospital, work on-call, total hours worked, direct patient hours, hours worked in community settings, and two self-nominated measures of practice complexity. 

### Analysis

Descriptive statistics were used to analyze longitudinal outcomes of geographic distributional and scope of practice for (i) wave 1 (2008), (ii) wave 6 (2013), and (iii) wave 10 (2017). Due to multiple fellowships, some doctors were counted in more than one category. Notably, all FARGPs also had a FRACGP qualification (a pre-set requirement), 5–10% of FACRRMs also had a FRACGP, while 25–35% of FARGPs also had a FACRRM. Secondary analyses limited respondents to only those who graduated from medical school after 1995, as a proxy for the cohort doing general practice training in the period of both the ACRRM and GradDip RurGP programs emerging, thus largely removing those awarded via full RPL. The discrete qualification categories were those (i) having a FACRRM; (ii) having a FARGP or GradDip RurGP (henceforth merged as ‘FARGP’); (iii) having a FRACGP and not having (i) or (ii); (iv) GPs not reporting any related qualification. Multiple logistic regression models were used to measure associations between these fellowships, other key characteristics, and the main geographic distribution outcomes. Sampling weights were used to adjust for survey non-response bias of key demographics. All analyses used Stata SE 15.1 for Windows (Stata Corp, College Station, TX, USA) and statistical significance was *p* < 0.05.

## 3. Results

In waves 1, 6, and 10 there were respectively 3930, 2936, and 3185 clinically active GPs who completed the MABEL survey. On average, in each wave there were 274 (8%) and 63 (2%) who, respectively, indicated they had either the FACRRM and/or FARGP qualifications.

FACRRMs were 75–83% male, compared with 50–65% for all other qualification groups (Table 3). Both FACRRMS and FARGPs were more likely to have a rural background (32–38%) than the other qualification groups (18–21%), but less likely to have been trained overseas (8–15%, compared with 22–31%). Reflective of their large RPL process, most FACRRMs were aged 50+. In contrast, most FARGPs were aged <50.

FACRRMs were mostly working in rural areas (75–84%) and approximately half were in the smaller communities (41–54%) (Table 4). Those with FARGPs were also mostly working in rural communities (56–67%), though proportionally fewer were in the smaller rural communities (25–41%). Around 50–60% of both FACRRMs and FARGPs were working in either Queensland or New South Wales, reflecting the largest rural populations. Amongst recent graduates, FACRRMs were moderately biased to working in Queensland (48%).

Both FACRRMs (26–31%) and FARGPs (29–34%) were more likely to be using advanced skills in their job, compared with those without these qualifications (14–26%) (Table 5). This was more pronounced for the four main procedural skills. However, FACRRMs and FARGPs were also more likely to have an advanced skill but not use it (13–26% vs 9–16%). Recent graduate FACRRMs (>1995) were more strongly related to maintaining their advanced skills than recent FARGP graduates. FACRRMs were most likely to work in a hospital setting and do on-call. FARGPs worked the longest hours per week, though both FARGPs and FACRRMs worked longer per week in other community settings. FACRRMs and FARGPs reported using less consultation support for complex patients, which is possibly reflective of their geographic distribution. FACRRMs reported mostly seeing patients with complex problems.

After adjusting for covariates (Table 6), FACRRMs were substantially more likely to be working in a rural area compared with those with standard qualifications (OR 8.7, 5.8–13.1), including when limited to graduates > 1995 (OR 9.6, 3.4–27.0). FACRRMs working rurally were significantly more likely to be working in smaller rural communities (OR 3.5, 2.3–5.3). FARGPs were also significantly more likely to work rurally (OR 4.2, 2.2–7.8). However, rural FARGPs were not more likely than those with standard qualifications to work in smaller rural communities (OR 1.1, 0.5–2.5). 

## 4. Discussion

This paper presents the first empirical evidence about the characteristics and geographic distribution of doctors related to rural general practice faculties compared with GPs who are not members of these faculties. It identifies GPs associated with both FACRRM and FARGP compared with GPs of standard qualifications. None significantly improved rural distribution and expanded the scope of practice. Though the faculties are structured in different ways and function relatively independently of each other, each faculty relates to members who work in rural communities at a broader scope of practice, with improved geographic distribution than those GPs who are not such faculty members. 

A key finding is that the stand-alone faculty that has a specific rural mission and delivers wholly rural training (FACRRM), relates to doctors of better distribution into smaller rural and isolated communities, as well as doctors who sustain practice of their advanced skills (working in areas like obstetrics atop of general practice, as rural generalists). These findings demonstrate the value of rural faculties as a professional hub for rural doctors, enabling rural-tailored training and professional support, as a critical strategy for growing and sustaining a skilled and geographically distributed primary care workforce. 

These data additionally provide a strong reminder that GPs associated with rural faculties remain a small minority of the trained general practice workforce, around 10% relative to 29% of Australians living rurally, and the 13% of Australians living in smaller rural and isolated communities (where rural generalist doctors are most indicated to be required). GPs working and living in large regional centers may not require specific professional training for their practice, and often have similar professional and personal experiences to colleagues in metropolitan areas [41]. However, strong growth of rural faculties might assist to address growing the skilled rural generalist workforce that is sorely needed in smaller rural towns. Further, most of the ACRRM fellows are older than 50 and will require replacement within 15–20 years. Currently, of around 1500 new general practice vocational training enrolments each year across Australia, there are approximately 150 FACRRM (10%) enrolments annually and around 85 FARGPs (6%). A February 2020 government announcement stated that ACRRM’s training intake will increase from 150 to 250 in future years, which is likely to be a welcome expansion.

Another potential source of faculty expansion to consider is to draw on the large proportion of international doctors, both those graduating domestically as international students or those migrating as graduates to Australia from their home country (OTDs). Each of these groups face Australian regulations that require up to 10 years of rural practice if they wish to access Medicare billing opportunities in Australia. Other research identifies that FGAMS have higher odds of working as a GP than local graduates, but decreased odds of working rurally [42]. Additionally, OTDs constitute a high and increasing proportion of GPs and other specialists in large and smaller rural communities [43]. Despite this, OTDs were seen to have considerably lower rural faculty membership, and there may be ways for current faculties to attract uptake of memberships by this group (and FGAMS), in order to encourage their experience of collegial and skills-supported rural practice. 

The FACRRM group are predominantly male (75–83%, or 60% for graduates >1995), despite the majority of Australia’s recent medical graduates being female (around 55–60%). This may relate to ACRRM’s relatively large initial recognition of prior learning process to grow the faculty at its initiation, but it may also reflect that female GPs are less likely to practice procedural skills and often desire more control of their working hours [44,45]. Flexible training options, supportive team practices with sufficient staff relief, salaried employment options, female-tailored continuing professional development topics, and robust social and professional network opportunities may be important strategies to attracting more females to this workforce [44,46,47]. Previous research has demonstrated the linkage between female GPs having children and relocations to more urban settings, with the same effect on males only occurring when the children are of secondary school age [48]. There appears to be a strong scope for rural faculties to play a role in accommodating the tailored employment and family needs of doctors.

Potentially related to their work locations, a higher proportion of rural faculty had a broader scope of work than standard qualified GPs. Notably, a higher proportion of FACRRMs who recently graduated (>1995) were using their advanced skills, whereas GPs mostly used four procedural types. This is likely capturing the strong association between the recent graduates working in Queensland where there is a specific state-based award, recognition, and remuneration for procedural rural generalist doctors [37]. This may also relate to FACRRMs, unlike FARGPs, compulsorily required to complete at least 12 months of training in smaller rural communities and 6 months of emergency medicine. Thus, their members may have greater confidence in working in more isolated communities requiring advanced skills. Maintaining advanced skills in procedural practice areas is likely to depend on matching training options to community need and ongoing job opportunities, availability of hospital departments with service gaps from other specialists, as well as employing adequate professional rewards and continuing learning support for advanced skill use [49]. 

A limitation of our study is that it likely has undercounted specific qualifications and advanced skills, as we relied on self-reported data. It is not possible to distinguish between incomplete (missing) data and genuine not applicable (missing) data. A further limitation of this study is that qualifications via RPL mostly cannot be distinguished from those related to completing training requirements. RPL was a major feature of ACRRM’s establishment and thus results of only more recent graduates are shown. This study presents a series of cross-sectional results, thus only associations rather than causality can be identified. A strength of this study is its use of national data, without focus on a single program; however, not all characteristics of the two rural faculty programs will readily match those of other countries.

## 5. Conclusions

This study demonstrates the value of different rural faculty models for building a skilled and qualified rural generalist GP workforce, over standard GP training. It highlights that rural faculties, whether as a standalone rural college (FACRRM) or embedded within an existing faculty (FARGP), reflect a common professional practice model. Both groups of rural faculty members related to a majority geographically distributed workforce (>50% in rural communities), practicing at a broader scope. FACRRM members, however, were more likely to work in smaller rural communities and retain use of their procedural skills. Our evidence suggests that rural faculties may better cater for a rural-ready primary care workforce with common professional practice models, providing potential gains for developing rural-specific networks, continuing professional development activities, and promoting recognition of rural practice. A key factor for future planning is maintaining objective data to evaluate further the critical design, progress, and outcomes of rural faculties against their specific missions to ensure that they remain fit for purpose. Expanding the utilization of rural faculties to sufficient capacity is likely to be a critical strategy for building and sustaining a primary care doctor workforce that meets the needs of rural communities.

## Figures and Tables

**Table 2 ijerph-17-04652-t002:** Training pathway to attaining either FARGP or FACCRM qualifications.

Pathway Component	FARGP (First 3 Years Are FRACGP) (All Rural or Part Rural/Metro)	FACRRM (All Rural)
Selection into general practice training ^#^	1350 places (RACGP) under the Australian General Practice Training (AGPT) program, enrolled with Regional Training Organization (RTO)	150 places (ACRRM), 3 pathways: AGPT—enrolled with Regional Training Organization Remote Vocational Training Scheme—typically remain in existing rural job, enroll with Regional Training Organization, remote supervision Independent pathway—enrolled with ACRRM only
Hospital training (core/foundation terms)	12 months, ‘rotations’ for: Adequate exposure to the discipline of medicine, surgery, emergency medicine and pediatrics	12 months, ‘rotations’ for: General surgery; general internal medicine; obstetrics and gynecology; pediatrics; anesthetics; emergency medicine.
General practice training terms	18 months: Accredited general practice training posts (rural or metropolitan)Guidance of RACGP-accredited supervisor	18 months: 6 months in a community primary care role 12 months living and working in small rural and remote practice (<15,000 communities), without ready access to specialist support
Hospital term (emergency/inpatient care)	Nil	6 months: Hospital care for admitted patients Emergency medicine in hospital emergency department
Extended skills term	6 months: Singular post or a combination of posts, develop an area of interest or address area of weakness (not necessarily an advanced skill)	N/A
Advanced skills training (AST)	FARGP enrollees only Minimum 12 months (some components can be concurrently completed during their FRACGP): 12 months in rural general practice (<50,000 communities) 6 months rural general practice community project (population health, with needs assessment) 12 months advanced skills training, one of Procedural: Anesthetics; obstetrics; emergency medicine; surgery Non-procedural: child health; mental health; aboriginal and Torres Strait Islander health; palliative care; adult internal medicine	Minimum 12 months (surgery requires 24 months), AST may be undertaken in one of the following disciplines: Procedural: Anesthetics; emergency medicine; obstetrics and gynecology; surgery Non-procedural: Aboriginal and Torres Strait Islander health; Academic Practice; adult internal medicine; mental health; pediatrics; population health; remote medicine
Supervision	Mix of FARGP, ACRRM fellows and other specialists	Mix of ACRRM fellows and other specialists
Professional development (PD, post fellowship)	Small rural-focused program (e.g., rural webinar series) overseen by the RACGP rural faculty Large range of PD events and courses available for all members, but mostly not rural specific	Large range of PD events, targeted at maintaining skills for *rural* general practice National annual conference for rural medicine Key voice for advocacy and policy reform at the national (and international) level in rural medicine

^#^ Control of general practice training has recently begun a transition phase from the Australian government’s AGPT, to ACRRM and RACGP from 2022 [38]. FACRRM: Fellowship of the Australian College of Rural and Remote Medicine; FARGP: Fellowship in Advanced Rural General Practice; RACGP: Royal Australian College of General Practitioners; ACRRM: Australian College of Rural and Remote Medicine.

**Table 3 ijerph-17-04652-t003:** Demographics of GP participants by fellowship group.

Characteristic	Wave 1 (2008), n = 3930				Wave 6 (2013), n = 2936				Wave 10 (2017), n = 3185			
	FACRRM	FARGP	FRACGP	None	FACRRM	FARGP	FRACGP	None	FACRRM	FARGP	FRACGP	None
Count ^	346	50	1718	1881	247	73	1513	1200	233	72	1667	1301
Male	83%	59%	55%	64%	75%	54%	54%	60%	81%	57%	50%	58%
Female	17%	41%	45%	36%	25%	46%	46%	40%	19%	43%	50%	42%
Metropolitan background	68%	63%	80%	82%	65%	68%	80%	80%	62%	67%	80%	79%
Rural background	32%	37%	20%	18%	35%	32%	20%	20%	38%	33%	20%	21%
AMG	85%	92%	78%	78%	87%	86%	70%	69%	90%	92%	76%	73%
OTD	15%	8%	22%	22%	13%	14%	30%	31%	10%	8%	24%	27%
<50 years	37%	94%	65%	34%	32%	78%	55%	44%	33%	74%	54%	47%
50+ years	63%	6%	35%	66%	68%	22%	45%	56%	67%	26%	46%	53%

^ Counted in two categories (FACRRM and FRACGP, or FACRRM and FARGP): Wave 1 = 65; Wave 6 = 97; Wave 10 = 88; AMG: Australian (or New Zealand) Medical Graduate; OTD: Overseas Trained Doctor; FACRRM: Fellowship of the Australian College of Rural and Remote Medicine; FARGP: Fellowship in Advanced Rural General Practice; FRACGP: Fellowship of the Royal Australian College of General Practitioners.

**Table 4 ijerph-17-04652-t004:** Geographic distribution of the Medicine in Australia: Balancing Employment and Life (MABEL) study. GP participants by fellowship group.

Geographic Region		Wave 1 (2008), n = 3930				Wave 6 (2013), n = 2936				Wave 10 (2017), n = 3185				Wave 10 (2017), Only Medical School Graduates > 1995, n = 1510			
**Australia’s Population (2016)**		**FACRRM**	**FARGP**	**FRACGP**	**None**	**FACRRM**	**FARGP**	**FRACGP**	**None**	**FACRRM**	**FARGP**	**FRACGP**	**None**	**FACRRM**	**FARGP**	**FRACGP**	**None**
Metropolitan (MMM 1)	71%	25%	35%	76%	77%	24%	33%	74%	69%	25%	38%	73%	71%	16%	44%	70%	60%
Large rural/regional (MMM 2–3)	16%	25%	33%	15%	15%	35%	26%	17%	18%	28%	33%	17%	18%	31%	31%	19%	24%
Small rural or isolated (MMM 4–7)	13%	50%	32%	9%	9%	41%	41%	9%	13%	48%	29%	10%	11%	53%	25%	11%	16%
**State Population—Rural Only (2016)**																	
Queensland	26%	18%	32%	22%	18%	26%	35%	24%	19%	29%	26%	21%	20%	48%	29%	21%	25%
New South Wales	28%	41%	28%	31%	37%	40%	21%	29%	37%	31%	30%	32%	34%	16%	24%	34%	33%
Victoria	20%	18%	19%	26%	24%	17%	21%	25%	23%	19%	21%	26%	25%	9%	12%	22%	22%
Other state	26%	22%	22%	20%	21%	18%	23%	22%	22%	21%	23%	21%	21%	27%	35%	23%	20%

Those with multiple fellowships were counted in each respective category; MMM: Modified Monash Model rurality classification; FACRRM: Fellowship of the Australian College of Rural and Remote Medicine; FARGP: Fellowship in Advanced Rural General Practice; FRACGP: Fellowship of the Royal Australian College of General Practitioners.

**Table 5 ijerph-17-04652-t005:** Scope of practice of MABEL GP participants by fellowship group.

Scope Measure	Wave 10 (2017), n = 3185				Wave 10 (2017), Only Medical School Graduates > 1995, n = 1510			
	FACRRM	FARGP	FRACGP	None	FACRRM	FARGP	FRACGP	None
Use any advanced skill	31%	34%	26%	21%	26%	29%	23%	14%
Use any procedural skill	20%	16%	7%	5%	24%	10%	6%	4%
Not use any advanced skill	26%	23%	16%	16%	13%	26%	11%	9%
Not use any procedural skill	23%	19%	11%	11%	12%	25%	8%	6%
Work in hospital	63%	50%	20%	16%	81%	48%	20%	17%
On call	64%	57%	28%	28%	70%	41%	25%	26%
Total work hours (mean)	44	41	37	36	47	43	36	37
Direct patient hours (mean)	33	30	30	30	35	34	30	31
Work community hours ^#^ (mean)	3.6	4.0	2.5	2.3	4.7	3.1	2.1	2.2
Consult with others ^1^	60%	66%	71%	74%	69%	63%	76%	83%
Complexity of patients ^2^	91%	78%	80%	79%	91%	70%	73%	74%

Those with multiple fellowships were counted in each respective category; ^#^ Aggregate of Community health center, Residential/aged care facility, Aboriginal health service; ^1^ “I normally consult with others in the practice about the management of patients with complex health and social problems”—% agree or strongly agree; ^2^ “The majority of my patients have complex health and social problems”—% agree or strongly agree; FACRRM: Fellowship of the Australian College of Rural and Remote Medicine; FARGP: Fellowship in Advanced Rural General Practice; FRACGP: Fellowship of the Royal Australian College of General Practitioners.

**Table 6 ijerph-17-04652-t006:** Multivariate logistic regression models of geographic distribution by fellowship group and other characteristics (Wave 10—2017, MABEL).

		All GPs		Rural GPs Only	
		Working Any Rural v Metropolitan, n = 2833	Working Any Rural v Metropolitan, Only Medical School Graduates > 1995, n = 1309	Working Small Rural (MMM4–7) v Large Rural (MMM 2–3), n = 1094	Working Small Rural (MMM4–7) v Large Rural (MMM 2–3), Only Medical School Graduates > 1995, n = 564
Reference Category	Characteristic	OR (95% CI)	OR (95% CI)	OR (95% CI)	OR (95% CI)
FRACGP	FACRRM	8.7 (5.8–13.1) **	9.6 (3.4–27.0) **	3.5 (2.3–5.3) **	3.6 (1.7–7.7) **
FRACGP	FARGP	4.2 (2.2–7.8) **	3.1 (1.4–6.8) **	1.1 (0.5–2.5)	1.2 (0.4–3.3)
FRACGP	None	1.2 (1.0–1.5) *	1.8 (1.4–2.4) **	1.2 (0.9–1.6)	1.2 (0.8–1.8)
Age <50	50+	0.6 (0.5–0.8) **	N/A	1.0 (0.7–1.3)	N/A
Male	Female	0.8 (0.7–1.0) *	0.8 (0.6–1.0)	0.8 (0.6–1.1)	0.9 (0.6–1.3)
AMG	OTD	1.4 (1.1–1.7) **	1.1 (0.8–1.5)	1.1 (0.8–1.5)	1.3 (0.8–2.0)
Metro background	Rural background	2.3 (1.9–2.8) **	2.6 (1.9–3.4) **	0.9 (0.7–1.2)	0.9 (0.6–1.4)

* *p* < 0.05; ** *p* < 0.01; Those with multiple fellowships were only counted in their first category (allocation order = FACRRM, FARGP, FRACGP, none); State was adjusted for in the model (coefficients are not shown as they largely reflect the population dispersion across Australia’s states); AMG: Australian (or New Zealand) Medical Graduate; OTD: Overseas Trained Doctor who gained basic medical qualifications an another country; FACRRM: Fellowship of the Australian College of Rural and Remote Medicine; FARGP: Fellowship in Advanced Rural General Practice; FRACGP: Fellowship of the Royal Australian College of General Practitioners.

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
