# Peer review of "Faculties to Support General Practitioners Working Rurally at Broader Scope: A National Cross-Sectional Study of Their Value"

_ijerph, 2020, doi:10.3390/ijerph17134652_

Round 1
Reviewer 1 Report
In line 173 - Are the values “3.5, 2.3-5.3” correct?
In table 3- I suggest some spacing between columns: “Wave 1”, “Wave 6” and “Wave 10”. Review the legend, adding the acronyms: FACRRM, FARGP and FRACGP.
In table 4 - I suggest add a line to: “Australia’s population (2016)”. I suggest some spacing or adding a column between columns: “Wave 1 (2018)”, “Wave 6 (2013)”, “Wave 10 (2017)” and “Wave 10 (2017), only medical school graduates > 1995”. Review the legend, adding the acronyms: FACRRM, FARGP and FRACGP.
In table 5 - I suggest some spacing between columns: “Wave 10 (2017)”, “Wave 10” and “Wave 10 (2017), only medical school graduates > 1995”. Review the legend, adding the acronyms: FACRRM, FARGP and FRACGP.
In table 6 - Review the legend, adding the acronyms: FACRRM, FARGP and FRACGP and the symbols “*” and “**”.
Author Response
Comment 1: In line 173 - Are the values “3.5, 2.3-5.3” correct?
Response: Thank you for picking up on this error in the text. It has been corrected to: 4.2, 2.2-7.8
Comment 2: In table 3- I suggest some spacing between columns: “Wave 1”, “Wave 6” and “Wave 10”. Review the legend, adding the acronyms: FACRRM, FARGP and FRACGP.
Response: Spacing has been added within the table, to better separate the Wave 1, 6 and 10 results. Acronyms have been added in the table footnote for FACRRM, FARGP and FRACGP.
Comment 3: In table 4 - I suggest add a line to: “Australia’s population (2016)”. I suggest some spacing or adding a column between columns: “Wave 1 (2018)”, “Wave 6 (2013)”, “Wave 10 (2017)” and “Wave 10 (2017), only medical school graduates > 1995”. Review the legend, adding the acronyms: FACRRM, FARGP and FRACGP.
Response: Australia’s population heading has been re-positioned, Spacing has been added within the table, to better separate the Wave 1, 6 and 10/10>1995 results. Acronyms have been added in the table footnote for FACRRM, FARGP and FRACGP.
Comment 4: In table 5 - I suggest some spacing between columns: “Wave 10 (2017)”, “Wave 10” and “Wave 10 (2017), only medical school graduates > 1995”. Review the legend, adding the acronyms: FACRRM, FARGP and FRACGP.
Response: Spacing has been added within the table, to better separate the Wave 10 & 10>1995 results. Acronyms have been added in the table footnote for FACRRM, FARGP and FRACGP.
Comment 5: In table 6 - Review the legend, adding the acronyms: FACRRM, FARGP and FRACGP and the symbols “*” and “**”.
Response: Acronyms have been added in the table footnote for FACRRM, FARGP and FRACGP. Explanations for * and ** have been added too.
Reviewer 2 Report
Some comments are suggested:
- The use of MeSH descriptors is recommended as keywords.
- The abstract should be structured according to IMRD and C clearly including the study design and avoiding abbreviations.
- You should try to improve the writing and use more points and followed to facilitate the reading of the introduction. For example, from line 29 to 42.
- The objective of the study should appear at the end of the introduction and now appears on line 41-42
- In the results, the number of subjects n = must be expressed together with the relative frequencies (%). Both in the text and in the tables
- Table 6 uses asterisks that I suppose refer to the significance p <0.05 or p <0.01, but it should be clarified in the legend of the table.
- All abbreviations must be clarified. Check
- Again in the discussion, it is advisable to break up and organize the paragraphs to improve their clarity and reading.
- In the discussion, greater impetus should be raised in the results found from the risk estimate (OR) and the comparison with other similar studies
- The conclusions could be improved by being more specific and referring to the objective of the study.
Author Response
Comment 1: The use of MeSH descriptors is recommended as keywords.
Response: Thankyou for this suggestion, we have modified our keywords to better align with MESH terms
Comment 2: The abstract should be structured according to IMRD and C clearly including the study design and avoiding abbreviations.
Response: We confirm that the abstract is structured in the IMRDC format, but without headings as this is what was requested in the author instructions. We have added the study design ‘a cross-sectional design’, in line with the title. We confirm that abbreviations for GP, FACRRM, FARGP and OR are now all expanded within the abstract.
Comment 3: You should try to improve the writing and use more points and followed to facilitate the reading of the introduction. For example, from line 29 to 42.
Response: We have edited some of the linkages between sentences in this paragraph and the following one, to improve its readability and clarity.
Comment 4: The objective of the study should appear at the end of the introduction and now appears on line 41-42
Response: We have shifted the objective to the end of paragraph 2 in the Introduction after we describe the intervention we are evaluating (rural faculties). We believe that the paper’s readability is improved by stating the objective early rather that having it at the end of the Introduction section.
Comment 5: In the results, the number of subjects n = must be expressed together with the relative frequencies (%). Both in the text and in the tables
Response: The overall n is given in the first paragraph of the Results. Additionally, we have added into Tables 3-6 the count (pre weights) in the header of each category. However, all results relating to Tables 3-6 (proportions, odds ratios) have been weighted to adjust for survey non-response, thus it is not helpful to provide n for all of these results.
Comment 6: Table 6 uses asterisks that I suppose refer to the significance p <0.05 or p <0.01, but it should be clarified in the legend of the table.
Response: Explanations for * and ** have been added to the table
Comment 7: All abbreviations must be clarified. Check
Response: Throughout the paper, we have checked that all acronyms are defined at their first usage and separately we have added footnotes in all tables
Comment 8: Again in the discussion, it is advisable to break up and organize the paragraphs to improve their clarity and reading.
Response: We have done further editing in the Discussion section, and broken some paragraphs, to improve its clarity.
Comment 9: In the discussion, greater impetus should be raised in the results found from the risk estimate (OR) and the comparison with other similar studies
Response: Essentially, there are no other similar studies and thus wider comparisons of our results (proportions, relative odds) are not possible. We made it clear this is the first study of this type.
Comment 10: The conclusions could be improved by being more specific and referring to the objective of the study.
Response: Our conclusion is closely related to the study’s objective, “to evaluate the benefit of rural faculties for supporting a more geographically distributed rural GP workforce, practicing at broader scope.” We have swapped the final two sentences, to achieve a more appropriate concluding remark from the study’s findings. Additional minor editing done in the Conclusion.
Reviewer 3 Report
The authors have clearly worked hard on this paper and for this they are to be congratulated.
This is a very interesting and useful contribution to the understanding of this area of medical practice in Australia.
It was interesting to note the (understandable) low rate of membership of the rural colleges. It is probably not the role of this paper to make what may be seen a 'politically sensitive' recomendations, but a requirement for OTD's to at least commence such qualification training would seem to be a logical outcome of this work.
Rural practice in all the professions has particular challenges, particularly for the professional's family as much as anything (Children to boarding school for advanced schooling, inability to be 'anonmymous' in the community, etc, etc)
I am not familiar with the survey instrument, but if there is any content relating to family issues other than basic demographics, it would be useful to include the more pertitent ones.
Otherwise I feel this is a worthwhile contribution to the canon.
Author Response
Comment 1: The authors have clearly worked hard on this paper and for this they are to be congratulated. This is a very interesting and useful contribution to the understanding of this area of medical practice in Australia.
Response: Thankyou for your feedback
Comment 2: It was interesting to note the (understandable) low rate of membership of the rural colleges. It is probably not the role of this paper to make what may be seen a 'politically sensitive' recomendations, but a requirement for OTD's to at least commence such qualification training would seem to be a logical outcome of this work.
Response: Thankyou for this interesting point. We have added a new paragraph and references into the Discussion, noting the disparity between the current low rural faculty uptake of OTDs (and locally graduating internationals – known as FGAMS) despite their high rural and/or GP presence, thus there would appear to be strong capacity to expand on their uptake.
Comment 3: Rural practice in all the professions has particular challenges, particularly for the professional's family as much as anything (Children to boarding school for advanced schooling, inability to be 'anonymous' in the community, etc)
Response: We agree, there is a large literature on the specific issues relating to recruitment and retention of rural doctors, but this was not a focus of this paper. Membership of rural faculties is primarily aimed at minimising some of the professional challenges. We further responded to this in comment 4.
Comment 4: I am not familiar with the survey instrument, but if there is any content relating to family issues other than basic demographics, it would be useful to include the more pertitent ones.
Response: Our previous research has investigated, using longitudinal panel data, specifically the linkage between family needs (having children of different ages, having a spouse in the workforce) and GPs staying or leaving rural practice. An additional 2 sentences has been added to the discussion on gender differences. It was not within scope of this paper to include analyses of non-professional factors, like family, as per our paper below which we have now referenced.
McGrail MR, O'Sullivan BG, Russell DJ. Family effects on the rurality of GP’s work location: A longitudinal panel study. Human Resources for Health. 2017;15:75.